# Adversarial Skill Chaining for Long-Horizon Robot Manipulation via Terminal State Regularization

**Youngwoon Lee**[1*]   **Joseph J. Lim**[1]   **Anima Anandkumar**[2,3]   **Yuke Zhu**[2,4]
[1]University of Southern California    [2]NVIDIA
[3]California Institute of Technology    [4]The University of Texas at Austin

**Abstract:** Skill chaining is a promising approach for synthesizing complex behaviors by sequentially combining previously learned skills. Yet, a naive composition of skills fails when a policy encounters a starting state never seen during its training. For successful skill chaining, prior approaches attempt to widen the policy's starting state distribution. However, these approaches require larger state distributions to be covered as more policies are sequenced, and thus are limited to short skill sequences. In this paper, we propose to chain multiple policies without excessively large initial state distributions by regularizing the terminal state distributions in an adversarial learning framework. We evaluate our approach on two complex long-horizon manipulation tasks of furniture assembly. Our results have shown that our method establishes the first model-free reinforcement learning algorithm to solve these tasks; whereas prior skill chaining approaches fail. The code and videos are available at https://clvrai.com/skill-chaining.

**Keywords:** Long-Horizon Manipulation, Skill Chaining, Reinforcement Learning

## 1 Introduction

Deep reinforcement learning (RL) presents a promising framework for learning impressive robot behaviors [1–4]. Yet, learning a complex long-horizon task using a single control policy is still challenging mainly due to its high computational costs and the exploration burdens of RL models [5]. A more practical solution is to decompose a whole task into smaller chunks of subtasks, learn a policy for each subtask, and sequentially execute the subtasks to accomplish the entire task [6–9].

However, naively executing one policy after another would fail when the subtask policy encounters a starting state never seen during training [6, 7, 9]. In other words, a terminal state of one subtask may fall outside of the set of starting states that the next subtask policy can handle, and thus fail to accomplish the subtask, as illustrated in Figure 1a. Especially in robot manipulation, complex interactions between a high-DoF robot and multiple objects could lead to a wide range of robot and object configurations, which are infeasible to be covered by a single policy [10]. Therefore, skill chaining with policies with limited capability is not trivial and requires adapting the policies to make them suitable for sequential execution.

To resolve the mismatch between the terminal state distribution (*i.e.* termination set) of one policy and the initial state distribution (*i.e.* initiation set) of the next policy, prior skill chaining approaches have attempted to learn to bring an agent to a suitable starting state [7], discover a chain of options [11, 12], jointly fine-tune policies to accommodate larger initiation sets that encompass terminal states of the preceding policy [6], or utilize modulated skills for smooth transition between skills [13–16, 9]. Although these approaches can widen the initiation sets to smoothly sequence several subtask policies, it quickly becomes infeasible as the larger initiation set often leads to an even larger termination set, which is cascaded along the chain of policies, as illustrated in Figure 1b.

Instead of enlarging the initiation set to encompass a termination set modelled as a simple Gaussian distribution [6], we propose to keep the termination set small and near the initiation set of the next policy. This can prevent the termination sets from becoming too large to be covered by the subsequent

---

*Work done during an internship at NVIDIA

5th Conference on Robot Learning (CoRL 2021), London, UK.

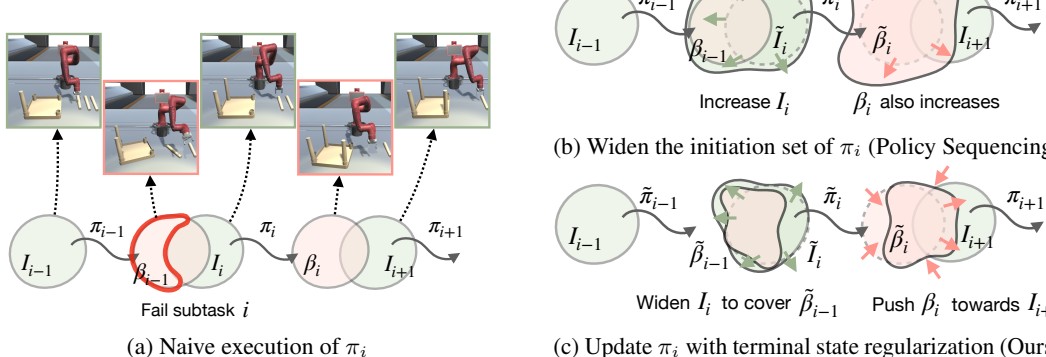

(a) Naive execution of $\pi_i$

(b) Widen the initiation set of $\pi_i$ (Policy Sequencing)

(c) Update $\pi_i$ with terminal state regularization (Ours)

Figure 1: We aim to solve a long-horizon task, *e.g.*, furniture assembly, using independently trained subtask policies. (a) Each subtask policy, $\pi_i$, works successfully only on its initiation set (green), $\mathcal{I}_i$, and results in its termination set (pink), $\beta_i$; thus, it fails when performed outside of $\mathcal{I}_i$ (red curve). (b) To enable sequencing policies, a subsequent policy $\pi_i$ needs to widen its initiation set to cover the termination set of the prior policy $\beta_{i-1}$. But this can result in an increase of its termination set $\beta_i$, which makes fine-tuning of the following policy $\pi_{i+1}$ even more challenging. This effect is exacerbated when more policies are chained together. (c) During fine-tuning of a policy $\pi_i$, we regularize the terminal state distribution $\beta_i$ to be close to the initiation set of the next policy $\mathcal{I}_{i+1}$. In contrast to the boundless increase of $\tilde{\beta}_i$ in (b), our approach effectively keeps the required initiation set small over the chain of policies with the terminal state regularization.

policies, especially when executing a long sequence of skills; thus, fine-tuning of subtask polices becomes more sample efficient. As a result, small changes in the initiation sets of subsequent policies are sufficient to successfully execute a series of subtask policies.

To this end, we devise an adversarial learning framework that learns an initiation set discriminator to distinguish the initiation set of the policy that follows from terminal states, and uses it as a regularization for encouraging the terminal states of a policy to be near the initiation set of the next policy. With this terminal state regularization, the pretrained subtask policies are iteratively fine-tuned to solve the subtask with the new initiation set while keep the terminal states of a policy small enough to be covered by the initiation set of the subsequent policy. As a result, our model is capable of chaining a sequence of closed-loop policies to accomplish a collection of multi-stage IKEA furniture assembly tasks [17] that require high-dimensional continuous control under contact-rich dynamics.

In summary, our contributions are threefold:

- We propose a novel adversarial skill chaining framework with *Terminal STAte Regularization*, T-STAR, for learning long-horizon and hierarchical manipulation tasks.

- We demonstrate that the proposed method can learn a long-horizon manipulation task, *furniture assembly* of two different types of furniture. Our terminal state regularization algorithms improves the success rate of the policy sequencing method from 0% to 56% for CHAIR_INGOLF and from 59% to 87% for TABLE_LACK. To our best knowledge, this is the first empirical result of a model-free RL method that solves these furniture assembly tasks without manual engineering.

- We present comprehensive comparisons with prior skill composition methods and qualitative visualizations to analyze our model performances.

## 2 Related Work

Deep reinforcement learning (RL) for continuous control [18–20] is an active research area. While some complex tasks can be solved based on a reward function, undesired behaviors often emerge [21] when tasks require several different primitive skills. Moreover, learning such complex tasks becomes computationally impractical as the tasks become long and complicated due to the credit assignment problem and the large exploration space.

Imitation learning aims to reduce this complexity of exploration and the difficulty of learning from reward signal. Behavioral cloning approaches [22–25] greedily imitate the expert policy and therefore suffer from accumulated errors, causing a drift away from states seen in the demonstrations. On the other hand, inverse reinforcement learning [26–28] and adversarial imitation learning approaches [29, 30] encourage the agent to imitate expert trajectories with a learned reward function, which can better handle the compounding errors. Specifically, generative adversarial imitation learning (GAIL) [29] and its variants [30, 31] show improved demonstration efficiency by training a discriminator to distinguish expert versus agent transitions and using the discriminator output as a reward for policy training. Although these imitation learning methods can learn simple locomotion behaviors [29] and a handful of short-horizon manipulation tasks [32, 31], these methods easily overfit to local optima and still suffer from temporal credit assignment and accumulated errors for long-horizon tasks.

Instead of learning an entire task using a single policy, we can tackle the task by decomposing it into easier and reusable subtasks. Hierarchical reinforcement learning does this by decomposing a task into a sequence of temporally extended macro actions. It often consists of one high-level policy and a set of low-level policies, such as in the options framework [33], in which the high-level policy decides which low-level policy to activate and the chosen low-level policy generates a sequence of atomic actions until the high-level policy switches it to another low-level policy. Options can be discovered without supervision [34–37, 11, 12], learned from data [38–40], meta-learned [41], pre-defined [7, 42–44], or attained through task structure supervision [45, 10]. For long-horizon planning capability, the high-level policy can be trained using traditional planning methods [46, 47].

To synthesize complex motor skills with a set of predefined skills, Lee et al. [7] learns to find smooth transitions between subtask policies. This method assumes the subtask policies are fixed, which makes learning such transitions feasible but, at the same time, leads to failure of smooth transition when an external state has to be changed or the end state of the prior subtask policy is too far away from the initiation set of the following subtask policy. On the other hand, Clegg et al. [6] proposes to sequentially improve subtask policies to cover the terminal states of previous subtask policies. While this method fails when the termination set of the prior policy becomes extremely large, our method prevents such boundless expansion of the termination set.

Closely related to our work, prior skill chaining methods [11, 12, 38] have proposed to discover a new option that ends with an initiation set of the previous option. With newly discovered options, the agent can reach the goal from more initial states. However, these methods have a similar issue with Clegg et al. [6] that discovering a new option requires a large enough initiation set of the previous option, which is cascaded along the chain of options.

Furniture assembly is a challenging robotics task requiring reliable 3D perception, high-level planning, and sophisticated control. Prior works [48–50] tackle this problem by manually programming the high-level plan and learning only a subset of low-level controls. In this paper, we propose to tackle the furniture assembly task in simulation [17] by learning all the low-level control skills and then effectively sequencing these skills.

## 3 Approach

In this paper, we aim to address the problem of chaining multiple policies for long-horizon complex manipulation tasks, especially in furniture assembly [17]. Sequentially executing skills often fails when a policy encounters a starting state (*i.e.* a terminal state of the preceding policy) never seen during its training. The subsequent policy can learn to address these new states, but this may require even larger states to be covered by the following policies. To chain multiple skills without requiring boundlessly large initiation sets of subtask policies, we introduce a novel adversarial skill chaining framework that constrains the terminal state distribution, as illustrated in Figure 2. Our approach first learns each subtask using subtask rewards and demonstrations (Section 3.2); it then adjusts the subtask policies to effectively chain them to complete the whole task via the terminal state regularization (Section 3.3).

### 3.1 Preliminaries

We formulate our learning problem as a Markov Decision Process [51] defined through a tuple $(\mathcal{S}, \mathcal{A}, R, P, \rho_0, \gamma)$ for the state space $\mathcal{S}$, action space $\mathcal{A}$, reward function $R(s, a, s')$, transition

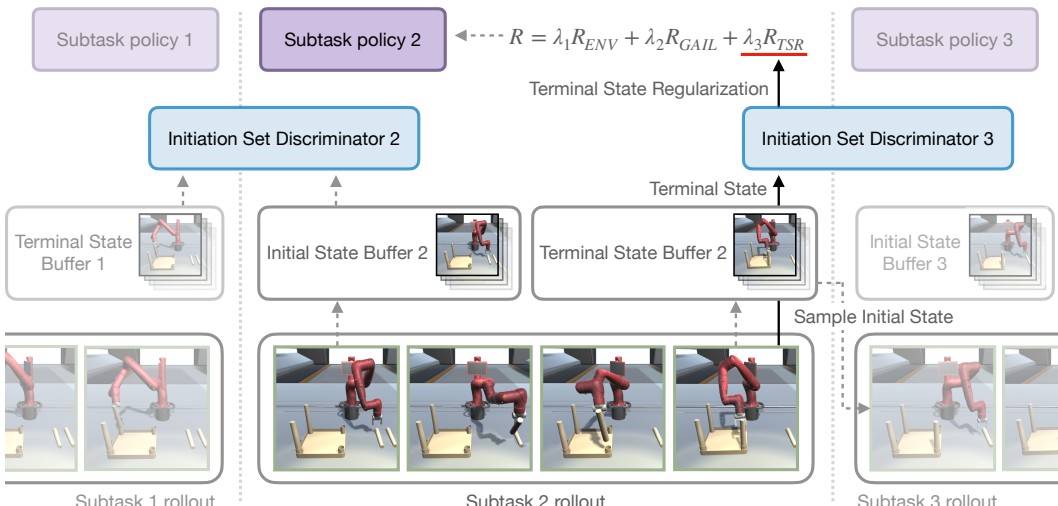

Figure 2: Our adversarial skill chaining framework regularizes the terminal state distribution to be close to the initiation set of the subsequent subtask. The *initiation set discriminator* models the initiation set distribution by discerning the initiation set and states in agent trajectories, while the policy learns to reach states close to the initiation set by augmenting the reward with the discriminator output, dubbed *terminal state regularization*. Our method jointly trains all policies and initiation set discriminators, pushing the termination set close to the initiation set of the next policy, which leads to smaller changes required for the policies that follow, especially effective in a long chain of skills.

distribution $P(s'|s, a)$, initial state distribution $\rho_0$, and discount factor $\gamma$. We define a policy $\pi : \mathcal{S} \to \mathcal{A}$ that maps from states to actions and correspondingly moves an agent to a new state according to the transition probabilities. The policy is trained to maximize the expected sum of discounted rewards $\mathbb{E}_{(s,a) \sim \pi} \left[ \sum_{t=0}^{T-1} \gamma^t R(s_t, a_t, s_{t+1}) \right]$, where $T$ is the episode horizon. Each policy comes with an initiation set $\mathcal{I} \subset \mathcal{S}$ and termination set $\beta \subset \mathcal{S}$, where the initiation set $\mathcal{I}$ contains all initial states that lead to successful execution of the policy and the termination set $\beta$ consists of all final states of successful executions. We assume the environment provides a success indicator of each subtask and this can be easily inferred from the final state, *e.g.*, two parts are aligned and the connect action is activated. In addition to the reward function, we assume the learner receives a fixed set of expert demonstrations, $\mathcal{D}^e = \{\tau_1^e, \ldots, \tau_N^e\}$, where a demonstration is a sequence of state-action pairs, $\tau_j^e = (s_0, a_0, \ldots, s_{T_j-1}, a_{T_j-1}, s_{T_j})$.

## 3.2 Learning Subtask Policies

To solve a complicated long-horizon task (*e.g.* assembling a table), we decompose the task into smaller subtasks (*e.g.* assembling a table leg to a table top), learn a policy $\pi_\theta^i$ for each subtask $\mathcal{M}_i$, and chain these subtask policies. Learning each subtask solely from reward signals is still challenging in robot manipulation due to a huge state space of the robot and objects to explore, and complex robot control and dynamics. Thus, for efficient exploration, we use an adversarial imitation learning approach, GAIL [29], which encourages the agent to stay near the expert trajectories using a learned reward by discerning expert and agent behaviors. Together with reinforcement learning, the policy can efficiently learn to solve the subtask even on states not covered by the demonstrations.

More specifically, we train each subtask policy $\pi_\theta^i$ on $R^i$ following the reward formulation proposed in [52, 53], which uses the weighted sum of the environment and GAIL rewards:

$$R^i(s_t, a_t, s_{t+1}; \phi) = \lambda_1 R^i_{ENV}(s_t, a_t, s_{t+1}) + \lambda_2 R^i_{GAIL}(s_t, a_t; \phi), \qquad (1)$$

where $R^i_{GAIL}(s_t, a_t; \phi) = 1 - 0.25 \cdot [f_\phi^i(s_t, a_t) - 1]^2$ is the predicted reward by the GAIL discriminator $f_\phi^i$ [53], and $\lambda_1$ and $\lambda_2$ are hyperparameters that balance between the reinforcement learning and imitation learning objectives. We found this reward formulation [53] works most stable among variants of GAIL [29] in our experiments thanks to its bounded reward between $[0, 1]$. The discriminator is trained using the following objective: $\min_{f_\phi^i} \mathbb{E}_{(s) \sim \mathcal{D}^e} \left[ (f_\phi^i(s) - 1)^2 \right] + \mathbb{E}_{(s) \sim \pi^i} \left[ (f_\phi^i(s) + 1)^2 \right]$.

---
**Algorithm 1** T-STAR: Skill chaining via terminal state regularization
---
**Require:** Expert demonstrations $\mathcal{D}_1^e, \ldots, \mathcal{D}_K^e$, subtask MDPs $\mathcal{M}_1, \ldots, \mathcal{M}_K$

1: Initialize subtask policies $\pi_\theta^1, \ldots, \pi_\theta^K$, GAIL discriminators $f_\phi^1, \ldots, f_\phi^K$, initiation set discriminators $D_\omega^1, \ldots, D_\omega^K$, initial state buffers $\mathcal{B}_\mathcal{I}^1, \ldots, \mathcal{B}_\mathcal{I}^K$, and terminal state buffers $\mathcal{B}_\beta^1, \ldots, \mathcal{B}_\beta^K$

2: **for** each subtask $i = 1, ..., K$ **do**
3:     **while** until convergence of $\pi_\theta^i$ **do**
4:         Rollout trajectories $\tau = (s_0, a_0, r_0, \ldots, s_T)$ with $\pi_\theta^i$
5:         Update $f_\phi^i$ with $\tau$ and $\tau^e \sim \mathcal{D}_i^e$             ▷ Train GAIL discriminator
6:         Update $\pi_\theta^i$ using $R^i(s_t, a_t, s_{t+1}; \phi)$ in Equation (1)      ▷ Train subtask policy
7:     **end while**
8: **end for**
9: **for** iteration $m = 0, 1, ..., M$ **do**
10:     **for** each subtask $i = 1, ..., K$ **do**
11:         Sample $s_0$ from environment or $\mathcal{B}_\beta^{i-1}$
12:         Rollout trajectories $\tau = (s_0, a_0, r_0, \ldots, s_T)$ with $\pi_\theta^i$
13:         **if** $\tau$ is successful **then**
14:             $\mathcal{B}_\mathcal{I}^i \leftarrow \mathcal{B}_\mathcal{I}^i \cup s_0, \mathcal{B}_\beta^i \leftarrow \mathcal{B}_\beta^i \cup s_T$   ▷ Collect initial and terminal states of successful trajectories
15:         **end if**
16:         Update $f_\phi^i$ with $\tau$ and $\tau^e \sim \mathcal{D}_i^e$            ▷ Fine-tune GAIL discriminator
17:         Update $D_\omega^i$ with $s_\beta \sim \mathcal{B}_\beta^{i-1}$ and $s_\mathcal{I} \sim \mathcal{B}_\mathcal{I}^i$       ▷ Train initiation set discriminator
18:         Update $\pi_\theta^i$ using $R^i(s_t, a_t, s_{t+1}; \phi, \omega)$ in Equation (3) ▷ Fine-tune subtask policy with terminal state regularization
19:     **end for**
20: **end for**
---

Due to computational limitations, training a subtask policy for all possible initial states is impractical, and hence it can cause failure on states unseen during training. Instead of indefinitely increasing the initiation set, we first train the policy on a limited set of initial states (*e.g.* predefined initial states with small noise), and later fine-tune the policy on the set of initial states required for skill chaining as described in the following section. This pretraining of subtask policies ensures the quality of the pretrained skills and makes the fine-tuning stage of our method easy and efficient.

### 3.3 Skill Chaining with Terminal State Regularization

Once subtask polices are acquired, one can sequentially execute the subtask policies to complete more complex tasks. However, naively executing the polices one-by-one would fail since the policies are not trained to be smoothly connected. As can be seen in Figure 1a, independently trained subtask policies only work on a limited range of initial states. Therefore, the execution of a policy $\pi_i$ fails on the terminal states of the preceding policy $\beta_{i-1}$ outside of its initiation set $\mathcal{I}_i$.

For successful sequential execution of $\pi_{i-1}$ and $\pi_i$, the terminal states of the preceding policy should be included in the initiation set of the subsequent policy, $\beta_{i-1} \subset \mathcal{I}_i$. This can be achieved either by widening the initiation set of the subsequent policy or by shifting the terminal state distribution of the preceding policy. However, in robot manipulation, the set of valid terminal states can be huge with freely located objects (*e.g.* a robot can mess up the workplace by moving or throwing other parts away). This issue is cascaded along the chain of policies, which leads to the boundlessly large initiation set required for policies, as illustrated in Figure 1b.

Therefore, a policy needs to not only increase the initiation set (*e.g.* assemble the table leg in diverse configurations) but also regularize the termination set to be bounded and close to the initiation set of the subsequent policies (*e.g.* keep the workplace organized), as described in Figure 1c. To this end, we devise an adversarial framework which jointly trains an *initiation set discriminator*, $D_\omega^i(s_t)$, to distinguish the terminal states of the preceding policy and the initiation set of the corresponding policy, and a policy to reach the initiation set of the subsequent policy with the guidance of the initiation set discriminator. We train the initiation set discriminator for each policy to minimize the following objective: $\mathcal{L}_i(\omega) = \mathbb{E}_{s_\mathcal{I} \sim \mathcal{I}_i} \left[ D_\omega^i(s_\mathcal{I}) - 1 \right]^2 + \mathbb{E}_{s_T \sim \beta_{i-1}} \left[ D_\omega^i(s_T) \right]^2$.

With the initiation set discriminator, we regularize the terminal state distribution of the policy by encouraging the policy to reach a terminal state close to the initiation set of the following policy. The *terminal state regularization* can be formulated as following:

$$R_{TSR}^i(s;\omega) = \mathbb{1}_{s\in\beta_i}D_\omega^{i+1}(s) \tag{2}$$

Then, we can rewrite the reward function with the terminal state regularization:

$$R^i(s_t,a_t,s_{t+1};\phi,\omega) = \lambda_1 R_{ENV}^i(s_t,a_t,s_{t+1}) + \lambda_2 R_{GAIL}^i(s_t,a_t;\phi) + \lambda_3 R_{TSR}^i(s_{t+1};\omega), \tag{3}$$

where $\lambda_3$ is a weighting factor for the terminal state regularization. The first two terms of this reward function guide a policy to accomplish the subtask while the terminal state regularization term forces the termination set to be closer to the initiation set of the following policy.

With this reward function incorporating the terminal state regularization, subtask policies and GAIL discriminators can be trained to cover unseen initial states while keeping the termination set closer to the initiation set of the next policy. Once subtask policies are updated, we collect terminal states and initiation sets with the updated policies, and train the initiation set discriminators. We alternate these procedures to smoothly chain subtask policies, as summarized in Algorithm 1, where changes of our algorithm with respect to Clegg et al. [6] are marked in red.

## 4 Experiments

In this paper, we propose a skill chaining approach with the terminal state regularization, which encourages to match the terminal state distribution of the prior skill with suitable starting states of the following skill. Through our experiments, we aim to verify our hypothesis that the policy sequencing fails due to unbounded terminal states, which is cascaded along the sequence of skills, and show the effectiveness of our framework on learning a long sequence of complex manipulation skills.

### 4.1 Baselines

We compare our method to the state-of-the-art prior works in reinforcement learning, imitation learning, and skill composition, which are listed below:

- **BC** [22] fits a policy to the demonstration actions with supervised learning.
- **PPO** [54] is a model-free on-policy RL method that learns a policy from the environment reward.
- **GAIL** [29] is an adversarial imitation learning approach with a discriminator trained to distinguish expert and agent state-action pairs $(s,a)$.
- **GAIL + PPO** uses both the environment and GAIL reward (Equation (1)) to optimize a policy.
- **SPiRL** [55] is a hierarchical RL approach that learns fixed-length skills and skill prior from the dataset, and then learns a downstream task using skill-prior-regularized RL.
- **Policy Sequencing** [6] first trains a policy for each subtask independently and finetunes the policies to cover the terminal states of the previous policy.
- **T-STAR (Ours)** learns subtask policies simultaneously, leading them to be smoothly connected using the terminal state regularization.

### 4.2 Tasks

We test our method and baselines with two furniture models, TABLE_LACK and CHIAR_INGOLF, from the IKEA furniture assembly environment [17] as illustrated in Figure 3:

- TABLE_LACK: Four table legs need to be picked up and aligned to the corners of the table top.
- CHAIR_INGOLF: Two chair supports and front legs need to be attached to the chair seat. Then, the chair seat needs to be attached to the chair back while avoiding collision to each other.

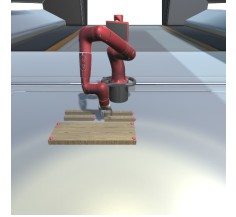 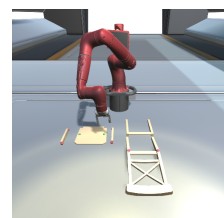

(a) TABLE_LACK      (b) CHIAR_INGOLF

Figure 3: Two furniture assembly tasks [17] consist of four subtasks (four table legs for TABLE_LACK; two seat supports, chair seat, and front legs for CHIAR_INGOLF).

In our experiments, we define a subtask as assembling one part to another; thus, we have *four* subtasks for each task. Subtasks are independently trained on the initial (object) states sampled from the environment with random noise ranging from $[-2cm, 2cm]$ and $[-3°, 3°]$ in the $(x, y)$-plane. This subtask decomposition is given, *i.e.*, the environment can be initialized for each subtask and the agent is informed whether the subtask is completed and successful.

For the robotic agent, we use the 7-DoF Rethink Sawyer robot operated via joint velocity control. For imitation learning, we collected 200 demonstrations for each furniture part assembly with a programmatic assembly policy. Each demonstration for single-part assembly consists of around 200-900 steps long due to the long-horizon nature of the task.

The observation space includes robot observations (29 dim), object observations (35 dim), and task phase information (8 dim). The object observations contain the positions (3 dim) and quaternions (4 dim) of all five furniture pieces in the scene. Once two parts are attached, the corresponding subtask is completed and the robot arm moves back to its initial pose in the center of the workplace. Our approach can in theory solve the task without resetting the robot; however, this resetting behavior effectively reduces the gap in robot states when switching skills and is available in most robots.

### 4.3 Results

The results in Table 1 show the average progress of the furniture assembly tasks across 200 testing episodes for 5 different seeds. Since each task consists of assembling furniture parts four times, completing one furniture part assembly amounts to task progress of 0.25. Even though the environment has small noises in furniture initialization, both BC [22] and GAIL [29] baselines move the robot arm near furniture pieces but struggle at picking up even one furniture piece. This shows the limitation of BC and GAIL in dealing with compounding errors in long-horizon tasks with the large state space and continuous action space. Similarly, the hierarchical skill-based learning approach, SPiRL, also strug-

Table 1: Average progress of the furniture assembly task. Each subtask amounts to 0.25 progress. Hence, 1 represents successful execution of all four subtasks while 0 means the agent does not achieve any subtask. Our method learns to complete all four subtasks in sequence and outperforms the policy sequencing baseline and standard RL and IL methods. We report the mean and standard deviation across 5 seeds.

|  | TABLE_LACK | CHAIR_INGOLF |
|---|---|---|
| BC [22] | $0.03 \pm 0.00$ | $0.04 \pm 0.01$ |
| PPO [54] | $0.09 \pm 0.11$ | $0.14 \pm 0.03$ |
| GAIL [29] | $0.00 \pm 0.00$ | $0.00 \pm 0.00$ |
| GAIL + PPO [52] | $0.21 \pm 0.11$ | $0.22 \pm 0.08$ |
| SPiRL [55] | $0.05 \pm 0.00$ | $0.03 \pm 0.00$ |
| Policy Sequencing [6] | $0.63 \pm 0.28$ | $0.77 \pm 0.12$ |
| T-STAR (Ours) | $0.90 \pm 0.07$ | $0.89 \pm 0.04$ |

gles at learning picking up a single furniture piece. This can be due to the insufficient amount of data to cover a long sequence of skills on the large state space, *e.g.*, many freely located objects. Moreover, due to the difficulty of exploration, the model-free RL baseline, PPO, rarely learns to assemble one part. On the other hand, the GAIL + PPO baseline can consistently learn one-part assembly, but cannot learn to assemble further parts due to the exploration challenge and temporal credit assignment problem. These baselines are trained for 200M environment steps (5M for off-policy SPiRL).

By utilizing pretrained subtask policies with GAIL + PPO (25M steps for each subtask), skill chaining approaches could achieve improved performance compared to the single-policy baselines. We train the policy sequencing baseline and our method for additional 100M steps, which requires in total 200M steps including the pretraining stage. The policy sequencing baseline [6] achieves 0.63 and 0.77 average task progress, whereas our method achieves 0.90 and 0.89 average task progress on TABLE_LACK and CHAIR_INGOLF, respectively. The performance gain of our method comes from the reduced discrepancy between the termination set and the initiation set of the next subtask thanks to the terminal state regularization.

We can observe this performance gain even clearer in the success rates. Our terminal state regularization improves the success rate of policy sequencing from 0% to 56% for CHAIR_INGOLF and from 59% to 87% for TABLE_LACK. We observe that with more skills to be chained, the success rate of the newly chained skill decreases, especially in CHAIR_INGOLF; the policy sequencing baseline learns to complete the first three subtasks, but fails to learn the last subtask due to excessively large and shifted initial state distribution, *i.e.*, terminal states of the preceding subtask.

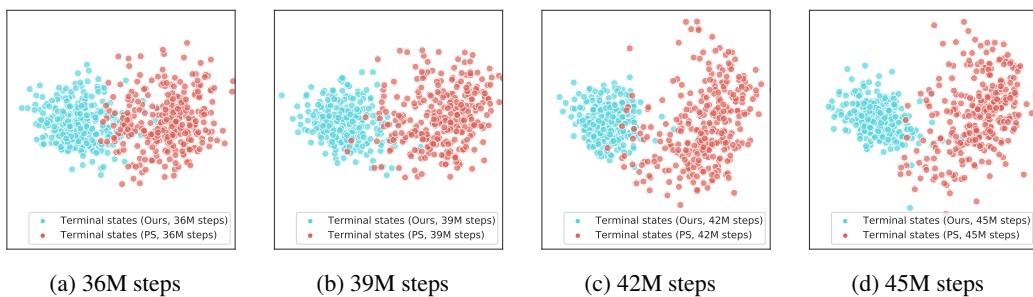

| (a) 36M steps | (b) 39M steps | (c) 42M steps | (d) 45M steps |

Figure 4: To demonstrate the benefit of our terminal state regularization, we visualize the changes in termination sets over training of the third subtask policy on CHAIR_INGOLF. We plot each terminal state by projecting its object configuration into 2D space using PCA. Through 36M to 45M training steps, both the policy sequencing baseline [6] and our method successfully learn to cover most terminal states from the second subtask. However, without regularization, the policy sequencing method (red) shows the increasing size of the termination set (*e.g.* spread over horizontally at 39M and vertically at 42M steps) as more initial states are covered by the policy. In contrast, in our approach (blue), the terminal state distribution is bounded, which shows that the terminal state regularization can effectively prevent the terminal state distribution diverging. This bounded termination set makes learning of the following skills efficient, and thus helps chaining a long sequence of skills.

## 4.4 Qualitative Results

To analyze the effect of the proposed terminal state regularization, we visualize the changes in termination sets over training. We first collect 300 terminal states of the third subtask policy on CHAIR_INGOLF both for our method and the policy sequencing baseline for 36M, 39M, 42M, and 45M training steps. Then, we apply PCA on the object state information in the terminal states and use the first two principal components to reduce the data dimension.

Figure 4 shows that our method effectively constrains the terminal state distribution of a subtask policy. Before 36M training steps, the policy cannot solve the third subtask due to the shifted initial state distribution by the second subtask policy. With additional training, at 42M and 45M training steps, the policy learns to solve the subtask on newly added initial states for both methods. From 36M to 45M steps of training, the termination set of the policy sequencing baseline (red) spreads out horizontally after 39M training steps and vertically after 42M steps. For successful skill chaining, this wide termination set has to be covered by the following policy, requiring a large change in the policy and potentially causing an even larger termination set. In contrast, the terminal state distribution of our method (blue) does not excessively increase but actually shrinks in this experiment, which leads to successful execution and efficient adaptation of subsequent subtask policies.

## 5 Conclusion

We propose T-STAR, a novel adversarial skill chaining framework that addresses the problem of the increasing size of initiation sets required for executing a long chain of manipulation skills. To prevent excessively large initiation sets to be learned, we regularize the terminal state distribution of a subtask policy to be close to the initiation set of the following subtask policy. Through terminal state regularization, our approach jointly trains all subtask policies to ensure that the final state of one policy is a good initial state for the policy that follows. We demonstrate the effectiveness of our approach on the challenging furniture assembly tasks, where prior skill chaining approaches fail. These results are promising and motivate future work on chaining more skills with diverse skill combinations to tackle complex long-horizon problems. Another interesting research direction is eliminating subtask supervision required in our work and discovering subtask decomposition from large data in a unsupervised learning manner. Finally, transferring our framework to real robot systems, which involves learning a vision-based policy, improving sample efficiency, and closing simulation-to-real gaps, is our definite future work.

## Acknowledgments

This work was initiated when Youngwoon Lee worked at NVIDIA Research as an intern. This research is also supported by the Annenberg Fellowship from USC and the Google Cloud Research Credits program with the award GCP19980904. We would like to thank Byron Boots for initial discussion, Jim Fan, De-An Huang, Christopher B. Choy, and NVIDIA AI Algorithms team for their insightful feedback, and the USC CLVR lab members for constructive feedback.

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
