# OpenReview forum: "Adversarial Skill Chaining for Long-Horizon Robot Manipulation via Terminal State Regularization"
_robot-learning.org/CoRL/2021/Conference — CoRL2021 Poster_

### Official Review · Reviewer_bN8e · 2021-07-19

**Originality:** Poor
**Technical Quality:** Good
**Clarity Of Presentation:** Very Good
**Impact:** 3

**Recommendation:**

Weak Accept: I recommend accepting the paper, but will not argue for my recommendation if the majority of other reviewers have a different opinion.

**Summary:**

The paper assumes that a segmentation for a long horizon task is given with individual demonstrations and reward function for each subskill, and aims to learn corresponding policies that, when executed sequentially, succeed in learning the overall task. To make sequential execution successful, the paper proposes to store for each subtask the initial and terminal states for each subskill, if it succeeded. A discriminator is then trained to classify between terminal states of the preceding subskill and successful initial states of the current subskill. The corresponding probability is provided as additional reward (on top of an RL and IL reward term) of the terminal state of the preceding subskill. The policies should thus learn to terminate in states where the subsequent skill succeeds. The approach is tested on a furniture assembly task and compared with a prior method [6] that instead of aiming to reach a good initial state for the next skill, aims to deal with the initial states of the previous skill.

**Issues:**

* The revision should discuss differences to prior work [A] and [B].

* Rebuttal and revision should make clear how the approach determines whether a subskill succeeded.

* Rebuttal and revision should clarify why 8 dimensions are used for encoding the phase of the skill.

* Rebuttal (at least) should explain why the classifier discriminates between terminal states of the last skill and successful initial states of the current skill, instead of discriminating between unsuccessful and successful initial states of the current skill.



**Reviewer Expertise:**

Good: General knowledge of the area

**Strengths And Weaknesses:**

Strengths
------------
* The paper addresses an important problem, namely, how to compose lower-level policies to solve long-horizon tasks.

* The presentation of the approach is mostly clear.

Weaknesses
------------------
* The discussion of prior work is insufficient. Namely, the paper completely misses the branch of research based on skill chaining by Konidaris et al., e.g. [A], [B]. These approaches also train an initiation set classifier and use them to train subskills to reach the initiation set of the next subskill. Whereas [A] considered a reinforcement learning setting, [B] considered learning from demonstrations.

* Especially, when relating the current work to these prior papers, I can not find a significant contribution. Providing additional reward to subskills for reaching initial states from which the subsequent policies succeed, seems to the main idea of the current submission, but this is idea is not novel ([A], [B]). The problem setting is a bit different from those works, because the current submission assumes that the overall task is already segmented and that subskill-specific reward functions are given, but I don't think that these simplifications increase the contribution.

* There is also no theoretical contribution as the presented approach is not based on any proofs or derivations.

* It is not clear how the proposed approach determines whether a subskill succeeds. Is this based on prior knowledge? Is this decision made on the respective state only, or is it maybe even non-Markovian?

* The experimental evaluation seems to be based on only one seed. The appendix contains a table (I assumed based on experiments that were performed after the paper submission deadline) for three seeds, which also contains confidence intervals. However, based on these confidence intervals, T-STAR might actually perform worse than Policy Sequencing (0.86-0.09 < 0.75+0.06). Also, even with three seeds, the experimental results seem insufficient.

* The "GAIL-reward" given in line 139 is very different from the reward used by GAIL. This difference needs to be discussed.

* The paper also uses a rather uncommon "AMP"-discriminator objectives and also an uncommon reward function for the terminal state regularization, which makes it difficult to interpret the effect of the additional reward term. Typical AIL reward functions would aim to minimize a divergence between the terminal state of subskill $i$ and the initiation set of the next subskill. I see that the chosen reward function in the range $[0,1]$ is easier to combine with the RL and IL reward objectives, but at least a short motivation for these design choices would be very important.

Minor
-------
* Why is the one-hot representation of the phase 8-dimensional, even though there are only four phases per task?

* Wouldn't it make more sense to train the classifier between successful and unsuccessful initial states?

* The paper is too repetitive regarding the difference (and benefits) between the current approach and [6]. Namely line 30-43, the caption of Fig. 1, 94-99, 153-161, are mainly repeating the same motivation. I would at least remove the paragraph 153-161.

* Caption should be more descriptive.

* in Figure 2, it would be good to show the "initial state buffer"-box also for subtask 1, and the "Terminal state buffer"-box also for subtask 3. For example, in the current Figure, it seems like "initiation Set Discriminator 3" is based on different types of data compared to "Initation Set Discriminator 2".

References
---------------

[A] Konidaris, George, and Andrew Barto. "Skill discovery in continuous reinforcement learning domains using skill chaining." Advances in neural information processing systems 22 (2009): 1015-1023.

[B] Konidaris, George, et al. "Robot learning from demonstration by constructing skill trees." The International Journal of Robotics Research 31.3 (2012): 360-375.


**Summary Of Recommendation:**

My recommendation is mainly based on the perceived lack of contribution. To me it seems like the paper proposes a hack (since no theoretical justification was given) for fine-tuning chained skills, that is not even significantly different from previous approaches. I also think that the discussion of prior work and experimental evaluation is insufficient.

---

> ### Author Response · Authors · 2021-08-30
> **Response to Reviewer bN8e (1/2)**
>
> Thank you for your constructive comments. We address your concerns in detail below and update our paper accordingly (changes marked in red).
>
> &nbsp;
>
> **The discussion of prior work is insufficient. When relating to the current work to these prior papers, I can not find a significant contribution**
>
> We thank the reviewers for pointing out these highly related works [a,b], which propose to discover and chain skills backward. We mistakenly dropped these references in our original submission but added them back to the updated manuscript (L91-95).
>
> First of all, these skill chaining approaches [a, b] have a similar issue with the policy sequencing baseline [6] but in the opposite direction. If options have small initiation sets, reaching to these initiation sets becomes harder, resulting in the failure of discovering a new option or an even smaller initiation set for a new option. Thus, discovering and chaining multiple options requires larger initiation sets for options but this is very challenging as we emphasized in the paper. In contrast, our adversarial skill chaining method resolves this issue with the terminal state regularization.
>
> As reviewers mentioned, both the prior skill chaining methods and our approach learn the initiation set classifiers. But, the prior works use the initiation set classifiers to __define and discover options__ (both for initiation and termination states) whereas our method uses the initiation set classifiers to __improve__ the subtask policies for skill chaining.
>
> Moreover, while the initiation set classifiers and options in the prior works are fixed once discovered, we iteratively and adversarially train the subtask policies and the initiation set classifiers, so that all skills can be jointly trained to accomplish the entire task.
>
> In summary, we identify a challenge in skill chaining, propose a novel adversarial skill chaining approach, and demonstrate our results on a complex long-horizon manipulation task, furniture assembly.
>
> &nbsp;
>
> **The experimental evaluation seems to be based on only one seed**
>
> As Reviewers noticed, we reported results with a single seed in Table 1 and with more seeds in Table 2 in appendix. To provide more reliable results, we ran two additional seeds and updated the results across **5 random seeds** in Table 1. As can be seen, the updated results are consistent with the original submission.
>
> Updated Table 1:
>
> | | table\_lack | chair\_ingolf |
> |-|-|-|
> | BC                       | 0.03 $\pm$ 0.00 | 0.04 $\pm$ 0.01 |
> | PPO                    | 0.09 $\pm$ 0.11 | 0.14 $\pm$ 0.03 |
> | GAIL                    | 0.00 $\pm$ 0.00 | 0.00 $\pm$ 0.00 |
> | GAIL+PPO          | 0.21 $\pm$ 0.11 | 0.22 $\pm$ 0.08 |
> | SPiRL                  | 0.05 $\pm$ 0.00 | 0.03 $\pm$ 0.00 |
> | PS                       | 0.63 $\pm$ 0.28 | 0.77 $\pm$ 0.12 |
> | T-STAR (Ours)    | 0.90 $\pm$ 0.07 | 0.89 $\pm$ 0.04 |
>
> Original Table 2:
>
> | | table\_lack | chair\_ingolf |
> |-|-|-|
> | BC                       | 0.02 $\pm$ 0.01 | 0.00 $\pm$ 0.00 |
> | PPO                    | 0.10 $\pm$ 0.02 | 0.13 $\pm$ 0.01 |
> | GAIL                    | 0.00 $\pm$ 0.00 | 0.00 $\pm$ 0.00 |
> | GAIL+PPO          | 0.27 $\pm$ 0.12 | 0.25 $\pm$ 0.11 |
> | SPiRL                  | 0.02 $\pm$ 0.01 | 0.01 $\pm$ 0.00 |
> | PS                       | 0.75 $\pm$ 0.06 | 0.68 $\pm$ 0.09 |
> | T-STAR (Ours)    | 0.86 $\pm$ 0.09 | 0.87 $\pm$ 0.07 |

---

> > ### Author Response · Authors · 2021-08-30
> > **Response to Reviewer bN8e (2/2)**
> >
> > **Short motivation for design choices on AMP and terminal state regularization**
> >
> > For the adversarial imitation learning design, we tried multiple variants of GAIL [Ho and Ermon. 2016] and found the formulation of AMP [Peng et al. 2020] works most stable in our experiments. The loss function of $f^i_\phi(s)$ is similar to the least-squares GAN formulation, which is widely used for GAN for stable training.
> >
> > As Reviewer bN8e mentioned, both the AMP reward and our terminal state regularization are bounded between [0, 1] by shifting, scaling, and clipping the discriminator output. This bounded reward makes the adversarial training much more stable whereas GAIL and AIRL’s reward functions sometimes provide overly large rewards, which makes training unstable. We included this discussion in the updated paper (L142-145).
> >
> > &nbsp;
> >
> > **Why is the one-hot representation of the phase 8-dimensional?**
> >
> > The one-hot vector represents the current phase (e.g. approach, grasp, pick up, or align), not the current subtask (e.g. assemble the first leg or second leg). Each subtask in the IKEA environment consists of 8 different phases, so the one-hot phase representation is 8-dimensional.
> >
> > &nbsp;
> >
> > **It is not clear how the proposed approach determines whether a subskill succeeds**
> >
> > We assume this is given from the environment and made based on the current state (Markovian). We clarified about the subtask success indicator in L124-126 and L220-222.
> >
> > &nbsp;
> >
> > **Wouldn’t it make more sense to train the classifier between successful and unsuccessful initial states?**
> >
> > Using successful and unsuccessful initial states to train the initiation set classifier can be an alternative to our method. However, testing whether each terminal state is successful or not for the following policy is costly. Moreover, some unsuccessful initial states can turn into successful states in the middle of policy training but these states cannot be identified later unless tested. Thus, we instead propose an adversarial approach which can encourage the terminal states to be close to the successful initial states.
> >
> > &nbsp;
> >
> > **The paper is repetitive regarding the difference (and benefits) between the current approach and [6]**
> >
> > We believe the motivation of our work is important to repeat but at the same time, we agree that we repeat it too many times in detail. Following the suggestion from Reviewer bN8e, we made L102-105 and L168-171 concise in the updated manuscript.
> >
> > &nbsp;
> >
> > **Caption should be more descriptive**
> >
> > Following your suggestion, we improved the captions, especially Figure 3 and Table 1.
> >
> > &nbsp;
> >
> > **In Figure 2, it would be good to show the “initial state buffer” box also for subtask 1, and the “terminal state buffer” box also for subtask 3.**
> >
> > Thank you for your suggestion! To avoid the confusion, we added “initial state buffer” for subtask 1 and “terminal state buffer for subtask 3 in the updated Figure 2.
> >
> >
> > &nbsp;
> >
> > **References**
> >
> > [a] Konidaris and Barto, Skill discovery in continuous reinforcement learning domains using skill chaining, NIPS 2009
> >
> > [b] Konidaris et al., Robot learning from demonstration by constructing skill trees, IJRR 2012
> >
> > &nbsp;
> >
> > Thanks again for your valuable feedback and spotting out our mistakes and typos. Please let us know if this addresses all your concerns and if there is any further concern or missing information that potentially prevent you from accepting this paper.

---

### Official Review · Reviewer_HKpX · 2021-07-23

**Originality:** Good
**Technical Quality:** Fair
**Clarity Of Presentation:** Good
**Impact:** 3

**Recommendation:**

Weak Reject: I recommend rejecting the paper, but will not argue for my recommendation if the majority of other reviewers have a different opinion.

**Summary:**

The paper proposes a novel learning objective for skill learning in the context of skill chaining.
Existing approaches to executing a sequence of skills fail when a subtask skill terminates outside a subsequent skill's initiation set.
To remedy this, existing approaches attempt to iteratively widen initiation sets to account for broad termination sets, but as such are limited to short skill sequences due to compounding error.
The paper proposes an adversarial learning objective for the termination condition such that the size of the termination set is regularized.
A discriminator is proposed which is trained to distinguish states from the termination set of one skill and the initiation set of the subsequent skill.
The discriminator is used as an additional term in the reward function to encourage subtask policies to reach the subsequent initiation set.


**Issues:**

- Please consider citing the original skill chaining paper from Konidaris and Barto. Termination and subsequent initiation sets are designed to be identical in this work.
- Are the subtask rewards distinct from the environment reward? If so, consider denoting subtask reward with R_{ENV}^i.
- Line 18 in the algorithm may have a typo? The loss in Equation (2) takes samples from the i-th initiation set and the (i-1)-th termination set, whereas it is written here with termination samples and initiation samples from the same skill execution.

**Reviewer Expertise:**

Good: General knowledge of the area

**Strengths And Weaknesses:**

- The adversarial formulation of learning the initiation set is clever.
- The main evaluation against existing methods is thorough, though the main table of results needs to be run across multiple seeds.
	Especially since the run reported in the main paper was (apparently) relatively lucky on the first task.
	Moreover, the improvement is relatively marginal.

- Relevant previous work is missing; esp. the original skill chaining paper.
- The theory is given as it pertains to skill chains, though skill trees also warrant consideration. Does the theory handle this case?

- Supplementary experimentation is lacking.
	It would be worth adding the initiation set to the visualization shown in Figure 4.
	This would provide compelling justification to the hypothesis presented regarding the mismatch between termination and subsequent initiation.
	I am also curious as to how something simple like (approximate) L2 regularization would do as a baseline.

- Lacks hardware results; understandable given COVID.

**Summary Of Recommendation:**

I'm recommending weak rejection. The idea has merit but isn't entirely original; the evaluation is a bit lacking. In particular, the main result should have been presented across multiple seeds, as in the appendix. Moreover, additional experimentation is warranted in the form of ablation to better understand the behavior of the proposed method, and the overview of existing work is lacking.

---

> ### Author Response · Authors · 2021-08-30
> **Response to Reviewer HKpX**
>
> Thank you for your constructive comments. We address your concerns in detail below and update our paper accordingly (changes marked in red).
>
> &nbsp;
>
> **The main table of results needs to be run across multiple seeds**
>
> As Reviewers noticed, we reported results with a single seed in Table 1 and with more seeds in Table 2 in appendix. To provide more reliable results, we ran two additional seeds and updated the results across **5 random seeds** in Table 1. As can be seen, the updated results are consistent with the original submission.
>
> Updated Table 1:
>
> | | table\_lack | chair\_ingolf |
> |-|-|-|
> | BC                       | 0.03 $\pm$ 0.00 | 0.04 $\pm$ 0.01 |
> | PPO                    | 0.09 $\pm$ 0.11 | 0.14 $\pm$ 0.03 |
> | GAIL                    | 0.00 $\pm$ 0.00 | 0.00 $\pm$ 0.00 |
> | GAIL+PPO          | 0.21 $\pm$ 0.11 | 0.22 $\pm$ 0.08 |
> | SPiRL                  | 0.05 $\pm$ 0.00 | 0.03 $\pm$ 0.00 |
> | PS                       | 0.63 $\pm$ 0.28 | 0.77 $\pm$ 0.12 |
> | T-STAR (Ours)    | 0.90 $\pm$ 0.07 | 0.89 $\pm$ 0.04 |
>
> Original Table 2:
>
> | | table\_lack | chair\_ingolf |
> |-|-|-|
> | BC                       | 0.02 $\pm$ 0.01 | 0.00 $\pm$ 0.00 |
> | PPO                    | 0.10 $\pm$ 0.02 | 0.13 $\pm$ 0.01 |
> | GAIL                    | 0.00 $\pm$ 0.00 | 0.00 $\pm$ 0.00 |
> | GAIL+PPO          | 0.27 $\pm$ 0.12 | 0.25 $\pm$ 0.11 |
> | SPiRL                  | 0.02 $\pm$ 0.01 | 0.01 $\pm$ 0.00 |
> | PS                       | 0.75 $\pm$ 0.06 | 0.68 $\pm$ 0.09 |
> | T-STAR (Ours)    | 0.86 $\pm$ 0.09 | 0.87 $\pm$ 0.07 |
>
> We also rewrote the captions and removed the confusion between the caption and results.
>
> &nbsp;
>
> **The original skill chaining paper is missing in related work. The idea has merit but isn’t entirely original**
>
> We thank the reviewers for pointing out these highly related works [a, b], which propose to discover and chain skills backward. We mistakenly dropped these references in our original submission but added them back to the updated manuscript (L91-95).
>
> First of all, these skill chaining approaches [a, b] have a similar issue with the policy sequencing baseline [6] but in the opposite direction. If options have small initiation sets, reaching to these initiation sets becomes harder, resulting in the failure of discovering a new option or an even smaller initiation set for a new option. Thus, discovering and chaining multiple options requires larger initiation sets for options but this is very challenging as we emphasized in the paper. In contrast, our adversarial skill chaining method resolves this issue with the terminal state regularization.
>
> As reviewers mentioned, both the prior skill chaining methods and our approach learn the initiation set classifiers. But, the prior works use the initiation set classifiers to __define and discover options__ (both for initiation and termination states) whereas our method uses the initiation set classifiers to __improve__ the subtask policies for skill chaining.
>
> Moreover, while the initiation set classifiers and options in the prior works are fixed once discovered, we iteratively and adversarially train the subtask policies and the initiation set classifiers, so that all skills can be jointly trained to accomplish the entire task.
>
> In summary, we identify a challenge in skill chaining, propose a novel adversarial skill chaining approach, and demonstrate our results on a complex long-horizon manipulation task, furniture assembly.
>
> &nbsp;
>
> **It would be worth adding the initiation set to Figure 4**
>
> We did not checkpoint the initiation sets so could not update Figure 4. We will include the initiation sets to Figure 4 in the revised version.
>
> &nbsp;
>
> **L2 regularization as a baseline**
>
> Thank you for suggesting a baseline that can show the contribution of our terminal state regularization. Due to the limited computing resources and time, we could not finish the experiments, but we will definitely include this in the revised version.
>
> &nbsp;
>
> **Are the subtask rewards distinct from the environment reward?**
>
> Thank you for your suggestion! In our experiments, the environment reward is the same across all subtasks (i.e. repeated four times); thus, we used the same reward function $R_{ENV}$ for all subtasks. However, we agree that a subtask-specific reward can be used in other environments, so we updated equations to use subtask-specific reward $R^i_{ENV}$.
>
> &nbsp;
>
> **Typo in L18, Algorithm 1**
>
> Thanks for spotting this! The terminal state should be sampled from $\mathcal{B}^{i-1}_\beta$. We fixed it in the updated paper.
>
>
> &nbsp;
>
> **References**
>
> [a] Konidaris and Barto, Skill discovery in continuous reinforcement learning domains using skill chaining, NIPS 2009
>
> [b] Konidaris et al., Robot learning from demonstration by constructing skill trees, IJRR 2012
>
>
> &nbsp;
>
> Thanks again for your valuable feedback and spotting out our mistakes and typos. Please let us know if there is any further concern that potentially prevents you from accepting this paper.

---

### Official Review · Reviewer_opMr · 2021-07-23

**Originality:** Fair
**Technical Quality:** Fair
**Clarity Of Presentation:** Very Good
**Impact:** 2

**Recommendation:**

Weak Accept: I recommend accepting the paper, but will not argue for my recommendation if the majority of other reviewers have a different opinion.

**Summary:**

This work proposes the algorithm T-STAR which approaches the problem of learning complex extended time-horizon skills through subtask skill chaining. T-STAR combines ideas from RL adversarial imitation learning to regularize the termination state distribution of a subtask policy to be close to the start state distribution of the next subtask. T-STAR builds on ideas in [6] by addressing some limitations in that method. The authors evaluate T-STAR on two furniture assembly tasks in simulation and show that a model-free RL method (T-STAR) is able to solve furniture assembly tasks without manual engineering.

**Issues:**

* Lack of robustness in results.
* Fair comparison of single policy baselines.
* Missing baseline from [7]
* Please also address the minor issues


**Reviewer Expertise:**

Good: General knowledge of the area

**Strengths And Weaknesses:**

**Strengths**

* The problem of learning complex long time-horizon skills efficiently is important and relevant to robot learning. The challenges in doing so are clearly identified and well explained. Finally the motivation for the proposed algorithm, T-STAR, is well grounded in the limitations of prior work (see section 3.3 which is especially clear).
* The proposed algorithm T-STAR is interesting and a novel approach to the skill chaining problem as far as I am aware. It also has the benefit of not requiring any additional policies to be trained over the sub-task policies.
* The qualitative results (section 4.4 and Figure 4) are a nice way of assessing if T-STAR is behaving as intended. It also visualizes the advantage of T-STAR over policy sequencing [6]. However I do have a question about the choice of time steps (see below)
* Since the authors are very clear about the differences between T-STAR and PS [6], the difference between PS and T-STAR in Table 1 quantifies the effectiveness of terminal state regularization. However I have a question about the robustness of the results (see below).
* Clarity of presentation is excellent. The paper is lucid and easy to follow. In particular, Figures 1 and 2 are very clear and do a great job of summarizing the main ideas and algorithm. The use of color highlighting in Algorithm 1 to highlight the differences with prior work is also useful.
* Inclusion of a mostly comprehensive set of baselines covering behavioral cloning, RL, imitation learning (adversarial), hierarchical RL and policy sequencing through initial state expansion.
* The implementation details (appendix), algorithm description (algorithm 1), and task description (section 4.2) appear to contain sufficient information for implementing T-STAR and reproducing the results.

**Weaknesses**

Lack of robustness in results
* Only a single score for each algorithm (both for task completion - Table 1 and success rate - lines 266-267) is presented in the main paper, and this appears to be from a single run. There is no discussion of variance in performance or why only a single result is presented.
* The appendix does present an additional result over 2 or 3 random seeds depending on the task, however there is a discrepancy between the results presented in the main paper (Table 1) and the results presented in the appendix (Table 2) with the results in the appendix on 3 of 4 key results being lower in the appendix.

Table 1:

PS 0.82 | 0.7

T-STAR 0.95 | 0.8

Table 2

PS 0.75 +/- 0.06 | 0.68 +/- 0.09

T-STAR 0.86 +/- 0.09 | 0.87 +/- 0.07

* It is not clear if Table 2 shows results with new seeds different to the seed used in Table 1 or if Table 2 includes the results in Table 1.
* It is also not clear why only 2 seeds are selected for TABLE_LACK but 3 for CHAIR_INGOLF
* There is a discrepancy between the caption for Table 2 (which implies that 3 seeds have been used for all tasks) and the text.
Looking at the results across Table 1 and Table 2, it appears that T-STAR performs significantly better on CHAIR_INGOLF but not on TABLE_LACK. However for the reasons described above this is difficult to assess.
* To address these robustness issues I suggest
    1. Unifying Table 1 from the main paper and Table 2 from the appendix and presenting the mean and std deviation across all random seeds in a single table in the main paper.
     2. Presenting the mean and standard deviation of the task success rate across all random seeds.
     3. Being clear about how many seeds were used in total for each task and reporting results averaged over all of these.
     4. Using the same number of seeds for both TABLE_LACK and CHAIR_INGOLF
     5. Not bolding results if the distributions overlap (e.g. Table 2: TABLE_LACK).

Single policy baselines do not appear to be a fair comparison with PS and T-STAR
* The single policy baselines BC, PPO, GAIL, GAIL+PPO are trained for 100M environment interactions, sPiRL is trained for 5M, whilst the PS and T-STAR results are trained for at least 200M interactions (not including the sub-task pre-training).

Whilst there is good coverage in baselines, two alternative approaches that are highlighted and cited by the authors as relevant are not included as baselines. One baseline is an important omission, the other is more minor.
* Transition policies from [7]. The authors reference this work frequently and it would be valuable to know how T-STAR compares to this approach. This approach directly addresses the policy sequencing problem, so without the inclusion of this baseline it is hard to assess the advantages of T-STAR over this method.
    1. The authors note that [7] is brittle to external state changes and can fail if the end state of a prior subtask policy is too far away from the initiation set of the following task. It would be valuable to see this assessed and compared with T-STAR experimentally on the furniture assembly task.
    2. [minor] The second approach is perhaps less directly relevant, so it is understandable why it is not included. However, a comparison with one modulated skills method e.g. one from [11 - 14, 9] would be interesting to see.

Minor
* Figure 4
     1. Why were timesteps 36M, 39M, and 42M steps chosen? Without any explanation it seems like these could have been cherry picked.
     2. It would be useful to see the same visualization evenly spaced throughout the fine tuning part of training, from 0 - 50M steps inclusive.
* Algorithm 1
    1. Lines 3 - 7: Is this repeated until convergence for each subtask or only carried out once?
     2. Line 5: Are the expert demonstrations (sampled from D^e) subtask specific or demonstrate the whole task. If the former then it appears that the subtask index i is missing. If the latter, then how are the relevant parts of the demonstration identified per subtask?
    3. Line10: What is the sampling ratio and how was it chosen?
* Table 1: Average progress over how many trials?
* Typos
    1. Line 28: “requires to adapting the policies to be suitable” -> “requires adapting the policies to make them suitable”
    2. Line 200: “an” -> “a”



**Summary Of Recommendation:**

Thank you for this submission. T-STAR is an interesting approach to an important problem in robot learning. T-STAR is well motivated, the paper is very clearly written and the main points are easy to understand. As-is however, it is not clear that T-STAR represents a significant improvement over PS or over the other relevant baseline from [7]. If the questions around robustness and baselines (described above) are addressed this would make a strong submission. In the future it would also be great to see results on a physical robot.

---

> ### Author Response · Authors · 2021-08-30
> **Response to Reviewer opMr**
>
> Thank you for your constructive comments. We address your concerns in detail below and update our paper accordingly (changes marked in red).
>
> &nbsp;
>
> **Lack of robustness in results**
>
> As Reviewers noticed, we reported results with a single seed in Table 1 and with more seeds in Table 2 in appendix. To provide more reliable results, we ran two additional seeds and updated the results across **5 random seeds** in Table 1. As can be seen, the updated results are consistent with the original submission.
>
> Updated Table 1:
>
> | | table\_lack | chair\_ingolf |
> |-|-|-|
> | BC                       | 0.03 $\pm$ 0.00 | 0.04 $\pm$ 0.01 |
> | PPO                    | 0.09 $\pm$ 0.11 | 0.14 $\pm$ 0.03 |
> | GAIL                    | 0.00 $\pm$ 0.00 | 0.00 $\pm$ 0.00 |
> | GAIL+PPO          | 0.21 $\pm$ 0.11 | 0.22 $\pm$ 0.08 |
> | SPiRL                  | 0.05 $\pm$ 0.00 | 0.03 $\pm$ 0.00 |
> | PS                       | 0.63 $\pm$ 0.28 | 0.77 $\pm$ 0.12 |
> | T-STAR (Ours)    | 0.90 $\pm$ 0.07 | 0.89 $\pm$ 0.04 |
>
> Original Table 2:
>
> | | table\_lack | chair\_ingolf |
> |-|-|-|
> | BC                       | 0.02 $\pm$ 0.01 | 0.00 $\pm$ 0.00 |
> | PPO                    | 0.10 $\pm$ 0.02 | 0.13 $\pm$ 0.01 |
> | GAIL                    | 0.00 $\pm$ 0.00 | 0.00 $\pm$ 0.00 |
> | GAIL+PPO          | 0.27 $\pm$ 0.12 | 0.25 $\pm$ 0.11 |
> | SPiRL                  | 0.02 $\pm$ 0.01 | 0.01 $\pm$ 0.00 |
> | PS                       | 0.75 $\pm$ 0.06 | 0.68 $\pm$ 0.09 |
> | T-STAR (Ours)    | 0.86 $\pm$ 0.09 | 0.87 $\pm$ 0.07 |
>
> &nbsp;
>
> **Single policy baselines do not appear to be a fair comparison with PS and T-STAR.**
>
> As reviewers mentioned, we used a different number of environment interactions for different methods as they _plateaued_ early on. In particular, we trained SPiRL for 5M iterations, which takes the similar amount of wall-clock time as 100M for PPO due to its slow off-policy RL update and did not show any progress in learning. Following reviewers’ suggestion, we ran all baselines, except for SPiRL, for the same amount of environment interactions (200M, which we used for PS and T-STAR in total) and updated Table 1.
>
> &nbsp;
>
> **Missing baseline Transition Policy [7]**
>
> Transition Policy [7] is learning an explicit policy that brings an agent to a good initial state for  the next skill. As also mentioned in [7], we found that Transition Policy struggles at chaining two skills when there are discrepancies in object poses. Thus, it mostly failed to connect the first two skills, resulting in 0.25 average progress. As per request of reviewers, we will include this baseline in our updated paper once we get multiple seeded results.
>
> &nbsp;
>
> **Why were timesteps 36M, 39M, 42M steps chosen in Figure 4?**
>
> In Figure 4, we visualize the distributions of terminal states when the baseline and our method started to show different learning curves. Due to the instability of training, we chose the evenly distributed checkpoints that achieve stable performance, which are 36M, 39M, 42M steps.
>
> As suggested by Reviewer opMr, we will include more visualizations throughout the training in Figure 4.
>
> &nbsp;
>
> **L3-7 in Algorithm 1: is this repeated until convergence for each subtask?**
>
> Thank you for spotting this! L3-7 should be repeated until convergence. We fixed it in the revised version.
>
> &nbsp;
>
> **L5 in Algorithm 1: are the expert demonstrations subtask specific or demonstrate the whole task?**
>
> In our experiments, we provide demonstrations for each subtask. Thus, $\mathcal{D}^e_i$ should be correct. We fixed it in the revised version.
>
> Our method can also work in the case where only demonstrations of the whole task are available. Our GAIL reward component simply encourages an agent to pass near the states seen in the demonstrations and the agent will never experience a state outside the subtask because the subtask policy will be terminated once it achieves the goal. Thus, learning from the whole task demonstrations will not hurt training of our method.
>
> &nbsp;
>
> **L10 in Algorithm 1: what is the sampling ratio and how was it chosen?**
>
> We sample an initial state from the environment for 20% and from the terminal state buffer for 80%. We use the low sampling rate for the environment’s initial states since the subtask policies are already good at solving the subtask but struggle at solving it from new initial states. We added this in the updated draft (appendix C.3).
>
> &nbsp;
>
> **How many trials are used for evaluation in Table 1?**
>
> We evaluated 200 episodes for each experiment.
>
>
> &nbsp;
>
> Thanks again for your valuable feedback and spotting out our mistakes and typos. Please let us know if this addresses all your concerns and if there is any further concern or missing information that potentially prevent you from accepting this paper.

---

> > ### Comment · Reviewer_opMr · 2021-09-02
> > **Response to authors**
> >
> > Thank you for responding to my comments so thoroughly and for your efforts to update the paper accordingly.
> >
> > The authors have comprehensively addressed all of my comments. In light of this I change my recommendation to weak accept.
> >
> > As I mentioned in my original review T-STAR is an interesting approach to an important problem in robot learning and is a valuable contribution to the research community. In light of the changes, the experimental evidence is now more convincing. I hope that the authors do include the Transition policy baseline in the main paper once all the seeded results are in and also update Figure 4 to include more visualizations throughout training.

---

### Official Review · Reviewer_8Fh5 · 2021-07-23

**Originality:** Very Good
**Technical Quality:** Good
**Clarity Of Presentation:** Very Good
**Impact:** 4

**Recommendation:**

Weak Accept: I recommend accepting the paper, but will not argue for my recommendation if the majority of other reviewers have a different opinion.

**Summary:**

The paper proposes a solution to long-horizon manipulation problem through chaining skills. The main contribution of the paper is an adversarial learning pipeline to bring the terminal states of one subtask close to the initial state of the upcoming one. The authors show that the proposed framework can significantly outperform IL, RL, IL+RL baselines, as well as an existing algorithm for skill chaining on two simulated IKEA furniture assemble tasks.

**Issues:**

See the "Strengths and Weaknesses" part.

**Reviewer Expertise:**

Good: General knowledge of the area

**Strengths And Weaknesses:**

Strengths:
- The presentation of the paper is very clear. The authors also provide well-made Figures to facilitate the understanding of the proposed solution, in particular, Figures 1 and 2.
- The research problem is well-motivated. The authors not only accurately point out the limitation of existing approaches, but also validate these intuitions in the experiments (Figure 4).
- The proposed solution of using adversarial learning is clean and neat. The algorithm proposed by the authors also seem very practical.
- The authors compare the proposed algorithms with various baselines including IL, RL, IL+RL, and anther skill-chaining approach. The authors also provide explanations on when and how the baseline fails.

Weaknesses:
- The authors did a thorough literature survey in the field of RL, IL, hierarchical RL, and skill chaining. However, it would be great if the authors can include existing work on learning for long-horizon robotic tasks, which is also relevant to the paper. A few examples here:
  - Driess et al. Learning Geometric Reasoning and Control for Long-Horizon Tasks from Visual Input. ICRA 2021
  - Kase et al. Transferable task execution from pixels through deep planning domain learning. ICRA 2020
- It would be great if the authors can expand the technical section (Section 3) a little to provide more technical details. In particular, it would be great if the authors can write out the loss function for $f_\phi^I$ rather than referring readers to [46]. I personally think it is important to also point out that the output range of $f_\phi^I$ is actually $[-1, 1]$ (rather than $[0,1]$ in more common formulations, including what the authors use in Eq. (2)). It actually took me a while to understand why there is a $0.25$ in for $R_{GAIL}$.
- Related to my previous point, it is interesting that the authors choose to use different adversarial formulations for the policy (IL) and the terminal states (TSR). The ranges of the discriminators are different ($[-1,1]$ for IL and $[0,1]$ for TSR); How the rewards are related to the discriminator is also different (quadratic reshaping for IL, while linear reshaping for TSR, both of them are different from the original formulation of $\log(\mathcal{D})$ in the original GAN/GAIL papers). Although the authors mentioned in the appendix that they observe the quadratic reward [46] works better. It would be great if the authors can provide more thorough discussion on these design choices and/or theoretical comparison of them.
- In the algorithm, the authors choose to pre-train the subtask policies (line 3-7) with a fixed set of initial states before the adversarial training. How does this pre-training stage help? It would be great if the authors can provide some intuitions and/or experimental results.
- Also in the algorithm, line 17-18, why is $\mathcal{D}_\omega^I$ updated after the policy $\pi_\theta^I$?
- Line 220, "Subtasks are independently trained on the initial states sampled from the environment with random noise ranging from [−2cm, 2cm] and [−3°, 3°] in the (x, y)-plane." Are the "states" here referred to the states of the furnitures? Why is the perturbation only on the (x,y)-plane?
- The fact that the authors reset the robot arm "back to its initial pose in the center of the workplace" after each subtask seems to shrink the set of terminal states quite a bit. Would the proposed method also generalize to when we don't reset the state of the robot after each subtask completion?

**Summary Of Recommendation:**

The paper solves a well-motivated problem for skill-chaining for long-horizon robot manipulation tasks. The authors accurately point out the limitation of the existing approaches, and also empirically validate these limitations in the experiments. The proposed solution seem technically sound and practical. The experiment validation is thorough. Though it would be great if the authors can justify some decision choices (see "Strengths and Weaknesses" part).

---

> ### Author Response · Authors · 2021-08-30
> **Response to Reviewer 8Fh5**
>
> Thank you for your constructive comments. We address your concerns in detail below and update our paper accordingly (changes marked in red).
>
> &nbsp;
>
> **Include existing work on learning for long-horizon robotic tasks.**
>
> Thank you for pointing out relevant works on solving long-horizon complex tasks. We included these works on tackling long-horizon tasks in our related work section (L85-86).
>
> &nbsp;
>
> **Detailed explanation in Section 3**
>
> As Reviewer 8Fh5 suggested, we included more technical details in the revised Section 3. For examples,
> * For the adversarial imitation learning design, we tried multiple variants of GAIL [Ho and Ermon. 2016] and found the formulation of AMP [Peng et al. 2020] works most stable in our experiments. The loss function of $f^i_\phi(s)$ is similar to the least-squares GAN formulation, which is widely used for GAN for stable training.
> * The reward is bounded between [0, 1] by shifting, scaling, and clipping the discriminator output. This bounded reward makes the adversarial training much more stable whereas GAIL and AIRL’s reward functions sometimes provide overly large rewards, which makes training unstable.
>
> &nbsp;
>
> **How does the pretraining stage help?**
>
> The pretraining stage can be integrated into the fine-tuning stage -- subtask policies can be jointly trained from scratch within our proposed framework. However, pretraining of subtask policies ensures the quality of the pretrained skills and makes the fine-tuning stage of Policy Sequencing [6] and our method easy and efficient. We described the pretraining procedure and this discussion in detail in the updated manuscript (L148-151).
>
> &nbsp;
>
> **Why is $D^i_\omega$ updated after the policy $\pi^i_\theta$?**
>
> Since the updates of $D^i_\omega$ and $\pi^i_\theta$ are iterative, the order of updating $D^i_\omega$ and $\pi^i_\theta$ is not very critical for overall training.
>
> &nbsp;
>
> **Are the “states” here referred to the states of the furniture? Why is the perturbation only on the (x,y)-plane?**
>
> Yes, the states in L220 refer to the positions and orientations of the furniture parts. The perturbation on this initial state is only performed on $(x,y)$-plane as randomizing $z$ position or rotation can lead to infeasible initial states, e.g., a part placed in the air or a part penetrating the table. Thanks for spotting this confusion. We clarified it in the updated draft.
>
> &nbsp;
>
> **Reset the robot arm after each subtask seems to shrink the set of terminal states. Would the proposed method also generalize to when we don’t reset the state of the robot after each subtask completion?**
>
> Skill chaining approaches require adjusting both robot and environment (object) states to make skill transition smooth. Our method can solve the task without resetting the robot. Nonetheless, in this paper, we chose to reduce the gap between terminal states of one subtask and initial states of the following subtask. This resetting behavior is not particularly artificial in the RL setting -- resetting a robot to its initial state is provided by most robots. We made this point clearer in the revised version (L230-232).
>
> &nbsp;
>
> Thanks again for your valuable feedback. Please let us know if this addresses all your concerns!

---

> > ### Comment · Reviewer_8Fh5 · 2021-09-03
> > **Thank you for your response**
> >
> > I would like to thank the authors for all the clarifications provided in the response and in the updated paper. I also see that the authors address some of other reviewers comments, such as providing the error bars for the experimental results. As mentioned by other reviewers, the paper can be strengthened by adding relevant baselines (as I understand, the authors have started the effort), and maybe test the proposed method on a real robot.
> >
> > I like the idea of the paper, and I think the presentation paper is quite clear, especially after the revision. Although I decide to keep my score as weak accept, I do think this paper is a solid contribution to CoRL, and recommend this paper being accepted (if only there is a score between strong and weak accept...)

---

### Meta-Review · Area_Chair_a3Vr · 2021-08-11

**Recommendation:** Accept (Poster)
**Confidence:** 5

**Metareview:**

The paper tackles the problem of long-horizon skill changing for manipulation tasks and proposes an adversarial learning algorithm - T-STAR - which brings the terminal states of one subtask close to the initial state of the upcoming one (avoiding the need for large initial state distributions required for baseline methods). The method is demonstrated to outperform various RL and IL baselines on two simulated IKEA furniture assembly tasks.

During the rebuttal phase, the authors greatly improved the quality of exposition and addressed the majority of reviewer concerns. All missing citations were added. The addition of std estimates (thanks to 5 seeded runs) shows that the empirical improvements are statistically significant. The experimental details were also made clearer. A majority of reviewers increased their recommended score and I agree with this sentiment. The paper is an interesting contribution to the thread of skill chaining in robot manipulation problems with strong empirical results and would be of benefit to the CoRL community.

---

> ### Author Response · Authors · 2021-08-30
> **Response to Meta Reviewer (1/2)**
>
> Thank you for summarizing all the important points from the reviewers. We first address common concerns from reviewers here and then respond to individual reviews. Following the suggestions from the reviewers, we have updated our paper accordingly (changes marked in red).
>
> &nbsp;
>
> **Missing citations: George Konidaris’ option discovery and skill chaining papers [HKpX, bN8e]**
>
> We thank the reviewers for pointing out these highly related works [a-c], which propose to discover and chain skills backward. We mistakenly dropped these references in our original submission but added them back to the updated manuscript (L30, L95-99).
>
> First of all, these skill chaining approaches [a-c] have a similar issue with the policy sequencing baseline [6] but in the opposite direction. In a complex environment, if options have small initiation sets, reaching to these initiation sets becomes harder, resulting in the failure of discovering a new option or an even smaller initiation set for a new option. Thus, discovering and chaining multiple options requires larger initiation sets for options but this is very challenging as we emphasized in the paper. In contrast, our adversarial skill chaining method tackles this issue with the terminal state regularization.
>
> As reviewers mentioned, both the prior skill chaining methods and our approach learn the initiation set classifiers. But, the prior works use the initiation set classifiers to __define and discover options__ (both for initiation and termination states) whereas our method uses the initiation set classifiers to __improve__ the subtask policies for skill chaining.
>
> Moreover, while the initiation set classifiers and options in the prior works are fixed once discovered, we iteratively and adversarially train the subtask policies and the initiation set classifiers, so that all skills can be jointly trained to accomplish the entire task.
>
> In summary, we identify a challenge in skill chaining, propose a novel adversarial skill chaining approach, and demonstrate our results on a complex long-horizon manipulation task, furniture assembly.
>
> &nbsp;
>
> **Results from a single seed in the main paper [opMr, HKpX]**
>
> As Reviewers noticed, we reported results with a single seed in Table 1 and with more seeds in Table 2 in appendix. To provide more reliable results, we ran two additional seeds and updated the results across **5 random seeds** in Table 1. As can be seen, the updated results are consistent with the original submission. Our method outperforms PS and shows more consistent results than PS.
>
> Updated Table 1:
>
> | | table\_lack | chair\_ingolf |
> |-|-|-|
> | BC                       | 0.03 $\pm$ 0.00 | 0.04 $\pm$ 0.01 |
> | PPO                    | 0.09 $\pm$ 0.11 | 0.14 $\pm$ 0.03 |
> | GAIL                    | 0.00 $\pm$ 0.00 | 0.00 $\pm$ 0.00 |
> | GAIL+PPO          | 0.21 $\pm$ 0.11 | 0.22 $\pm$ 0.08 |
> | SPiRL                  | 0.05 $\pm$ 0.00 | 0.03 $\pm$ 0.00 |
> | PS                       | 0.63 $\pm$ 0.28 | 0.77 $\pm$ 0.12 |
> | T-STAR (Ours)    | 0.90 $\pm$ 0.07 | 0.89 $\pm$ 0.04 |
>
> Original Table 2:
>
> | | table\_lack | chair\_ingolf |
> |-|-|-|
> | BC                       | 0.02 $\pm$ 0.01 | 0.00 $\pm$ 0.00 |
> | PPO                    | 0.10 $\pm$ 0.02 | 0.13 $\pm$ 0.01 |
> | GAIL                    | 0.00 $\pm$ 0.00 | 0.00 $\pm$ 0.00 |
> | GAIL+PPO          | 0.27 $\pm$ 0.12 | 0.25 $\pm$ 0.11 |
> | SPiRL                  | 0.02 $\pm$ 0.01 | 0.01 $\pm$ 0.00 |
> | PS                       | 0.75 $\pm$ 0.06 | 0.68 $\pm$ 0.09 |
> | T-STAR (Ours)    | 0.86 $\pm$ 0.09 | 0.87 $\pm$ 0.07 |
>
> &nbsp;
>
> **The empirical gain in improvements seems marginal [HKpX]**
>
> Although the intervals of the results have overlaps between Policy Sequencing (PS) and ours, our method clearly shows higher average progress than PS. More importantly, our method accomplishes sequencing all four sub-skills more than 87% and 55.5% for the table and chair assembly while PS achieves 59% and 0%, respectively.

---

> > ### Author Response · Authors · 2021-08-30
> > **Response to Meta Reviewer (2/2)**
> >
> > **How does the pretraining stage help? [8Fh5]**
> >
> > The pretraining stage can be integrated into the fine-tuning stage -- subtask policies can be jointly trained from scratch within our proposed framework. However, pretraining of subtask policies ensures the quality of the pretrained skills and makes the fine-tuning stage of Policy Sequencing [6] and our method easy and efficient. We described the pretraining procedure and this discussion in the updated manuscript (L148-151).
> >
> > &nbsp;
> >
> > **Missing baseline Transition Policy [7] [opMr]**
> >
> > Transition Policy [7] is learning an explicit policy that brings an agent to a good initial state for  the next skill. As also mentioned in [7], we found that Transition Policy struggles at chaining two skills when there are discrepancies in object poses. Thus, it mostly failed to connect the first two skills, resulting in 0.25 average progress. As per request of reviewers, we will include this baseline in our updated paper once we get multiple seeded results.
> >
> > &nbsp;
> >
> > **Inconsistency between the number of environment interactions between methods compared [opMr]**
> >
> > As reviewers mentioned, we used a different number of environment interactions for different methods as they _plateaued_ early on. In particular, we trained SPiRL for 5M iterations, which takes the similar amount of wall-clock time as 100M for PPO due to its slow off-policy RL update and did not show any progress in learning. Following reviewers’ suggestion, we ran all baselines, except for SPiRL, for the same amount of environment interactions (200M, which we used for PS and T-STAR) and updated Table 1.
> >
> > &nbsp;
> >
> > **Lack of real-world robotic experiments**
> >
> > We discuss the applicability of our approach to real robotic systems in detail in appendix, Section D. Our approach has multiple benefits over alternative RL approaches like “vanilla” SAC in terms of applicability to real robot systems. Most importantly, it learns the subtask policies independently, which has a shorter horizon and is easy to reuse. As a result it requires a lot fewer costly real-world environment interactions than alternative RL approaches. Its ability to leverage demonstrations can further improve sample efficiency. We evaluate our system on complex, long-horizon manipulation tasks with realistic high-DoF robotic agents. Thus, we believe that the proposed algorithm has promise for real-world application, although we were not able to directly test on real robot systems due to the circumstances of the pandemic.
> >
> > &nbsp;
> >
> > **References**
> >
> > [a] Konidaris and Barto, Skill discovery in continuous reinforcement learning domains using skill chaining, NIPS 2009
> >
> > [b] Konidaris et al., Robot learning from demonstration by constructing skill trees, IJRR 2012
> >
> > [c] Bagaria and Konidaris, Option discovery using deep skill chaining, ICLR 2020

---

### Decision · Program_Chairs · 2021-09-13

**Decision:**

Accept (Poster)

**Comment:**

The paper tackles the problem of long-horizon skill changing for manipulation tasks and proposes an adversarial learning algorithm - T-STAR - which brings the terminal states of one subtask close to the initial state of the upcoming one (avoiding the need for large initial state distributions required for baseline methods). The method is demonstrated to outperform various RL and IL baselines on two simulated IKEA furniture assembly tasks.

During the rebuttal phase, the authors greatly improved the quality of exposition and addressed the majority of reviewer concerns. All missing citations were added. The addition of std estimates (thanks to 5 seeded runs) shows that the empirical improvements are statistically significant. The experimental details were also made clearer. A majority of reviewers increased their recommended score and I agree with this sentiment. The paper is an interesting contribution to the thread of skill chaining in robot manipulation problems with strong empirical results and would be of benefit to the CoRL community.